# Immunoscore Combining CD8, FoxP3, and CD68-Positive Cells Density and Distribution Predicts the Prognosis of Head and Neck Cancer Patients

**DOI:** 10.3390/cells11132050

**Published:** 2022-06-28

**Authors:** Sonia Furgiuele, Géraldine Descamps, Jerome R. Lechien, Didier Dequanter, Fabrice Journe, Sven Saussez

**Affiliations:** 1Department of Human Anatomy and Experimental Oncology, Faculty of Medicine, Research Institute for Health Sciences and Technology, University of Mons (UMONS), Avenue du Champ de Mars, 8, B7000 Mons, Belgium; sonia.furgiuele@umons.ac.be (S.F.); geraldine.descamps@umons.ac.be (G.D.); fabrice.journe@umons.ac.be (F.J.); 2Department of Otolaryngology and Head and Neck Surgery, CHU Saint-Pierre, 1000 Brussels, Belgium; jerome.lechien@umons.ac.be (J.R.L.); didier.dequanter@pandora.be (D.D.); 3Laboratory of Clinical and Experimental Oncology, Institute Jules Bordet, Université Libre de Bruxelles (ULB), 1000 Brussels, Belgium

**Keywords:** immunoscore, CD8, FoxP3, CD68, head and neck cancer, prognosis, tumor microenvironment, tumor infiltration

## Abstract

We assessed immune cell infiltrates to develop an immunoscore for prognosis and to investigate its correlation with the clinical data of patients with head and neck cancer. CD8, FoxP3, and CD68 markers were evaluated by immunohistochemistry in 258 carcinoma samples and positive cells were counted in stromal and intra-tumoral compartments. The RStudio software was used to assess optimal cut-offs to divide the population according to survival while the prognostic value was established by using Kaplan–Meier curves and Cox regression models for each immune marker alone and in combination. We found with univariate analysis that the infiltration of immune cells in both compartments was predictive for recurrence-free survival and overall survival. Multivariate analysis revealed that CD8+ density was an independent prognostic marker. Additionally, the combination of CD8, FoxP3, and CD68 in an immunoscore provided a significant association with overall survival (*p* = 0.002, HR = 9.87). Such an immunoscore stayed significant (*p* = 0.018, HR = 11.17) in a multivariate analysis in comparison to tumor stage and histological grade, which had lower prognostic values. Altogether, our analysis indicated that CD8, FoxP3, and CD68 immunoscore was a strong, independent, and significant prognostic marker that could be introduced into the landscape of current tools to improve the clinical management of head and neck cancer patients.

## 1. Introduction

Head and neck squamous cell carcinomas (HNSCC) are among the most prevalent cancers worldwide, setting them in the sixth place [1]. In Belgium, their incidences are higher and such cancers arise at the fourth position in men [2]. Despite advances in therapeutic approaches, the mortality rate has remained relatively constant in recent years, with a 5-year survival rate around 50% and recurrences occurring in 40–60% of treated patients [3]. This poor response to treatment can be explained in part by a late diagnosis and a lack of efficient drugs in the case of tumor recurrence. However, it appears that the cell composition of the tumor microenvironment (TME) is likely to influence patient outcome [4]. The well-known risk factors of HNSCC are the consumption of alcohol and tobacco, as well as infection with the human papillomavirus (HPV), which is known to be associated with a better prognosis for the youngest patients with oropharyngeal carcinoma [5]. In this context, several studies suggest that HPV+ patients with HNSCC have a specific TME that may influence the response to treatment [6,7,8,9].

Macrophages, and more specifically tumor-associated macrophages (TAMs), are the most abundant cells within the TME and they are able to stimulate regulatory T lymphocyte (Treg) cells to switch to a pro-tumor environment [10,11]. Regarding macrophages, we previously showed that CD68+ infiltration arises during HNSCC progression in the intra-tumoral (IT) compartment and is associated with the tumor stage. We also highlighted that CD68+ recruitment is higher in HPV+ patients than in HPV− ones. Moreover, a high infiltration of CD68+ cells was related to a short recurrence-free survival (RFS) as well as a short overall survival (OS) [8].

Immune surveillance is also governed by the tumor-infiltrating T lymphocytes (TILs) [12]. Among them, the CD8+ T lymphocytes act specifically on the cancer cells in order to eliminate them [13]. In HNSCC, a high density of CD8+ is correlated with a good prognosis [14]. Concerning Treg cells, which are characterized by the transcription factor forkhead box P3 (FoxP3) [15], we previously showed that FoxP3+ Treg infiltration increased during HNSCC progression (from dysplasia to carcinoma) and that tumors with high Treg infiltration were associated with a longer RFS and OS [9,16].

In advanced HNSCC, the gold standard treatment remains concomitant chemoradiotherapy (CCR), but the emergence of immunotherapy over the last years has changed the landscape of HNSCC treatment [17,18]. However, using anti-cancer drugs such as immunotherapy is challenging due to the heterogeneity of the TME composition [19]. Importantly, HPV status is now included in the tumor-node-metastasis (TNM) staging system [20], indicating the importance of additional prognostic information to propose the most appropriate treatment for patients. Currently, there is no immune-based classification of head and neck cancer. However, the evaluation of immune cell recruitment to classify HNSCC patients in different immunologic subgroups depending on the TME composition could be helpful to improve patient prognosis. As such, a new classification model for colorectal cancers (CRC) has been already established by Galon et al. and validated in clinical trials [21]. It is based on the evaluation of the density of CD3+ and CD8+ T cells in two distinct compartments (tumor and invasive margins) resulting in a high or low immunoscore (IS). This consensus IS has been shown to outperform the TNM system in predicting the survival and recurrence of patients with stage I-III CRC. Specifically, a high IS was significantly associated with a longer survival and a lower risk of recurrence. Given the reproducibility and accuracy of this IS, it is now easily applied routinely to guide the treatment regimen of these patients [22]. The value of IS has also been highlighted in other cancers, such as cervical cancer [23], hepatocellular carcinoma [24], melanoma [25], gastric cancer [26], pancreatic cancer [27], and lung cancer [28], and it is starting to draw attention in HNSCC. On this point, Zhang et al. recently showed the interest of CD3+ and CD8+ cell infiltration scoring in combination with the TNM staging system in HNSCC patients [29], and our group reported that a high stromal FoxP3+ T cell number combined with tumor stage improved prognosis in HNSCC patients [16].

In this study, we propose an immune signature based on CD8+, FoxP3+, and CD68+ count in IT and/or stromal (ST) compartments in a large clinical series of 258 patients with HNSCC. The IS is compared to tumor stage and histological grade using multivariate analyses.

## 2. Materials and Methods

### 2.1. Patients and Clinical Data

A total of 258 patients presenting HNSCC were enrolled in our study. Table 1 describes the clinicopathological characteristics, treatment, and follow-up data. Formalin-fixed paraffin-embedded (FFPE) specimens obtained after surgical resection at Saint-Pierre Hospital (Brussels, Belgium), Jules Bordet Institute (Brussels, Belgium), EpiCURA Baudour Hospital (Baudour, Belgium), and CHU Sart-Tilman (Liège, Belgium) between 2002 and 2019 were used for immunohistochemical labeling. This retrospective study has been approved by the Institutional Review Board (Jules Bordet Institute, number CE2319).

### 2.2. HPV Status

DNA extractions were performed on FFPE tissue (10 slices of 5 µm) with the QIAmp DNA Mini Kit (Qiagen, Benelux, Antwerp, Belgium), according to the manufacturer’s recommended protocol. The HPV status of some patients was established by qPCR at the Algemeen Medisch Laboratorium (Antwerp, Belgium). Moreover, to determine the transcriptional activity of HPV, p16 immunostaining was performed using the recommended mouse monoclonal antibody (CINtec p16, clone E6H4, Ventana, Tucson, AZ, USA) on an automated immunostainer (BenchMark Roche, Ventana, AZ, USA at the Jules Bordet Institute (Brussels, Belgium) as previously described [7]. The expression of p16 was defined as positive only when both the nucleus and cytoplasm were stained and over 70% of the tumor cells were stained.

### 2.3. Immunohistochemistry

The 5 μm thick slices of HNSCC were deparaffinized in toluene and rehydrated in a graded series of alcohols, then peroxidase was blocked using H2O2 and finally slices were rinsed with water for 7 min. Antigen retrieval was processed by immersing the samples in 10% EDTA/H2O or in 10% citrate/H2O followed by heating in a pressure cooker or in a microwave (buffer and timing are dependent on antibodies, see Appendix A). Non-specific sites were blocked with 0.5% caseine for 15 min. Slices were incubated with primary antibody (anti-human CD68 monoclonal mouse, dilution 1:200, and anti-human CD8 monoclonal mouse, dilution 1:200, both from Dako (Uden, The Netherlands)) for 1 h at room temperature or overnight at 4 °C. Kit PowerVision Poly-HRP IgG were used for the second antibody. For FoxP3 immunostaining, the anti-human FoxP3 monoclonal mouse (dilution 1:200, from Invitrogen, MA, USA) was used and the detection of this primary antibody was performed with the CSAII kit (Dako, Uden, The Netherlands). For each immunohistochemistry, tonsil tissue was used as positive (and negative (no primary antibody)) controls.

### 2.4. Calculation of an IS

The number of each immune cell type was counted in 5 fields in the IT and ST compartments with an Axio-Cam MRC5 optical microscope (Zeiss, Hallbergmoos, Germany) at 400× magnification by two investigators (S.F. and G.D.). The mean of each counting was calculated for each patient and normalized in 1 mm^2^ area. For each marker in IT and ST, the cut-off value giving the best separation between two groups (HR and p for OS) was evaluated using the RStudio software. Then, if the mean density of the 5 fields was greater than the cut-off, the case was considered as “high” and if it was lower, the case was considered as “low”. Based on such cut-offs, the prognostic value of each immune marker was examined regarding RFS and OS. From these analyzes, an IS was defined combining the most significant immune markers.

### 2.5. Statistical Analyses

The optimal cut-off points of the population for each immuno-biomarker were calculated by using RStudio software (package from Cutoff Finder web application). Collected data were analyzed with IBM SPSS software (version 23) (IBM, Ehningen, Germany). RFS and OS analyses were performed using Kaplan–Meier curves. Univariate and multivariate Cox regression models were applied to calculate the hazard ratio (HR), 95% confidence interval and significance. *p*-values < 0.05 were statistically significant.

## 3. Results

### 3.1. Correlations between Clinical Characteristics and RFS or OS

Our clinical series included a total of 258 HNSCC patients, among which 177 (68.6%) were men and 81 (31.4%) were women, with a median age of 61 years old (range, 29–90). Among these patients, 104 patients presented tumor recurrence and 102 died. The clinicopathologic characteristics are provided in Table 1.

We evaluated the association between tumor stage, histological grade, or risk factors with RFS or OS. Cox regression models highlighted that among such parameters only tumor stage correlated with RFS and OS, and histological grade with OS. Evaluating Kaplan–Meier survival curves, patients with tumor stage I-II were associated with both a longer RFS (*p* = 0.041) and OS (*p* = 0.001) compared to patients with tumor stage III–IV. Moreover, well differentiated tumors were also associated with a longer OS (*p* = 0.029) compared to poorly differentiated ones (Figure 1).

### 3.2. Immune Cell Density and Patient Survival

In the HNSCC surgical specimens, immune cells were detected by using specific antibodies against CD8, FoxP3, and CD68 within the ST and the IT compartments (Figure 2A–C). Lymphocytes T (CD8+), Treg (FoxP3+), and macrophages (CD68+) were counted in five random fields (magnification 400×) in both ST and IT compartments (Figure 2D–I).

Cut-offs were calculated using RStudio software regarding optimal HR and *p*-values for OS and patients were classified as expressing a low or high density of immune cells. The cut-off values were 308.3 cells/mm^2^ (CD8, ST), 295.9 cells/mm^2^ (CD8, IT), 328.7 cells/mm^2^ (FoxP3, ST), 63.5 cells/mm^2^ (FoxP3, IT), 792.8 cells/mm^2^ (CD68, ST), and 122.1 cells/mm^2^ (CD68, IT). Additionally, univariate and multivariate analyses (Cox regression) were performed for the three immuno-markers CD8, FoxP3, and CD68 in the two compartments for RFS and OS (Table 2). Multivariate analysis showed that the CD8+ cell density was a strong and independent prognostic marker.

Using these cut-offs, Kaplan–Meier curves were established for each immune cell in each compartment for RFS and OS. Regarding the ST compartment, a longer RFS was significantly associated with a high FoxP3+ cell density, while a longer OS correlated with a low CD8+ and a high FoxP3+ cell density (Figure 3). In ST, the CD68+ cell density did not correlate with RFS or OS. In the IT compartment, a high CD8+, a high FoxP3+, and a low CD68+ cell density were significantly linked to a longer RFS as well as a longer OS (Figure 4).

### 3.3. IS and Patient Survival

The Figure 5A describes how we calculated our IS. Each tumor of the patients was categorized into a low (Lo) or high (Hi) density for each immune cell in each tumor region according to the cut-off values. Depending on the type of immune cells and the tumor compartment, the Lo and Hi classes were associated to the blue and red groups corresponding to the 0 and 1 scores, respectively. The IS was created by adding the individual score (0/1) of each marker, which was significant for OS. Based on univariate analyses (Table 2), we included CD8 ST/IT, FoxP3 ST/IT, and CD68 IT in the IS. The CD68+ cell density in the ST did not correlate with the RFS or OS and were therefore not included in the IS. Thus, the scoring system ranged from 0 to 5. Kaplan–Meier curves were drawn for each value of the IS and a cut-off discriminating good and poor patient prognosis was chosen at three (Appendix A). Thus, each patient was classified in the blue group (good prognosis, low immune score < 3, n = 23) or in the red group (poor prognosis, high immune score > 3, n = 97).

Very significant differences were observed between the blue and red IS. Kaplan–Meier curves using our IS showed a significant correlation for RFS (*p* = 0.007) and OS (*p* = 0.002) (Figure 5B,C, respectively) with a high IS (>3) being associated with a shorter RFS and OS, compared with a low IS (<3) that was associated with a better prognosis for RFS and OS. Nevertheless, we are aware of the heterogeneity of our clinical cohort, but, as a first approach, we wished to obtain a global immunoscore for all head and neck cancers combined. However, we performed univariate and multivariate Cox regression models for CD8, FoxP3, and CD68 in the two compartments (ST and IT), for RFS and OS, and for six patient subtypes of our cohort. Then, we evaluated our IS in each population (Appendix A).

Finally, we performed univariate and multivariate Cox regression analyses to compare our IS with the conventional tumor stage and histological grade. Our IS correlated more significantly, and with a greater separation of the two groups, regarding OS (*p* = 0.002, HR = 9.87) compared to tumor stage (*p* = 0.005, HR = 1.91) and histological grade (*p* = 0.029, HR = 0.62) (Table 3). Multivariate analyses revealed that the IS was the only parameter associated with a strong and independent prognosis value.

## 4. Discussion

To the best of our knowledge, this study assessed, for the first time, the abundance and distribution of innate and adaptive cellular elements according to CD8 T cells, FoxP3, Treg, and CD68 macrophages in a series of 258 patients with HNSCC in order to define a more global immune contexture. Then, we investigated their potential prognostic value separately and in combination to stratify patients using a low IS corresponding to a longer RFS and OS, whereas a high IS was related to a poorer prognosis. Our results confirm that the establishment of an IS has a higher prognostic value than those of the TNM staging system and histological grade.

For many years, clinical research around head and neck cancers has been constantly asking for new prognostic biomarkers to better guide patient management. Given the complexity of the interactions between immune infiltrates within the TME, the tumor must no longer be considered as a single entity but must be studied in relation to its microenvironment and the host immune response in order to bring clinical relevance and value in determining the tumor progression and the patient prognosis. As such, clinicians and researchers have been interested in the infiltration of immune cells in several types of cancer. The most widely studied and established IS is for colorectal cancers, so it is now used by clinicians, along with TNM stage, as a predictive and prognostic marker. In fact, many studies have highlighted that CD8+/CD3+ cytotoxic T cells infiltration in tumors or in invasive margins will classify patients with a low or high IS that can predict patient prognosis [21,30,31]. In gastric cancer, IS with eight immune cell types (dendritic cells, mast cells, activated CD4+ T cells, effector memory CD8+ T cells, type-17 T helper cells, CD56+ natural killer cells, activated B cells, and memory B cells) revealed that patients with a low IS are associated with a longer DFS and OS. In the same study, they have demonstrated that the combination of the eight IS markers with the TNM was superior to the TNM stage alone, pointing out the interest of using an immune signature [32]. Moreover, in cervical cancer, increasing CD45RA+/CD45RO+ and decreasing CCL20+/CCR6+ expression correlated with neoplasia severity [33]. Additionally, lung cancer is also reported as another immune infiltrated cancer. Besides, Feng et al. demonstrated that completely resected stage IIIA(N2) non-small cell lung cancer with high CD45RO+ and CD8+ cells infiltration in the tumor center and invasive margin can predict longer distant metastasis-free survival and OS [34]. Concerning head and neck cancer, in most cases, massive infiltration of CD8+ T lymphocytes in HPV− as well as in HPV+ oropharyngeal carcinomas correlates positively with patient prognosis [35,36,37,38,39,40,41]. CD8+ TILs have also been established as an independent prognostic marker in patients diagnosed with oropharyngeal squamous cell carcinoma [42]. Recently, Echarti et al. confirmed these findings by quantifying CD8+ and FoxP3+ T lymphocytes in epithelial and ST compartments [43]. The relationship between tumor infiltrating Treg and patient prognosis has been evaluated in many malignancies and is associated with a different prognostic response according to the sites of primary cancer. The prognostic significance of FoxP3+ Tregs has been extensively studied [44] and high Treg cell recruitment is reported to be correlated with poor prognosis in breast, liver, pancreatic, ovarian, cervical, and renal cancer [45,46,47,48,49,50], whereas it may also correlate with a longer survival as demonstrated in colorectal, bladder cancer, HNSCC, and lymphoma [9,51,52,53,54]. The meta-analysis of Shang et al. compares 76 datasets that highlight the prognostic role of FoxP3+ in 17 cancer types [44]. For HNSCC, Treg infiltration is highly controversial with conflicting results. While several studies underline the deleterious impact of massive infiltration by Tregs on prognosis, we and others have already showed the opposite [53,55]. Indeed, we previously reported that FoxP3+ infiltration was associated with a longer RFS and OS of patients suffering from HNSSC [9,16,44]. The concomitance that is often observed between Treg and cytotoxic T infiltrates could tip the balance towards a favorable immune orientation and consequently participate in the good paradoxical prognostic value of this population. In addition, HNSCCs are heterogeneous tumors located in certain anatomical sites rich in lymphoid tissue such as oropharynx, which may explain a greater recruitment of Treg in this site. Importantly, a crucial point of the debate was elucidated in 2016 when Saito et al. demonstrated the existence of two populations of FoxP3+ Tregs in colorectal cancers [56]. The first one was the immunosuppressive Foxp3^high^-expressing cells classically associated with a poor prognosis and the second was characterized by non-suppressive capacities and the absence of the expression of CD45 (FoxP3^low^ CD45RA-), the naïve T cell marker. These two populations are significantly correlated with opposing prognoses in colorectal cancer. While the first is associated with a poor prognosis, the infiltration of FoxP3^low^, which secrete pro-inflammatory cytokines such as IL-12 and TGFβ, is characteristic of better patient survival [56]. In addition, we suggest another hypothesis that could explain the infiltration of FoxP3+ Tregs in head and neck cancers. Indeed, they grow in a septic environment in contact with a resident microbiota, just like colon cancers that are also associated with a better prognosis when the density of Treg is high. This microbiota interacts and may modulate the host oral immune cells and such alterations in Treg functions have already been observed in oral infections [57,58]. Moreover, studies demonstrated the protective effects of oral FoxP3+ Treg in some local infections [59]. Given the crosstalk between these immunosuppressive regulatory cells and the cytotoxic lymphocytes, which are the anticancer mediators of the immune system, it seems crucial to consider these two entities in combination in order to better understand the reasons for the contradictory results reported in the literature.

Macrophages also constitute an important partner in innate and adaptive inflammatory responses. Beyond the binary classification of pro-inflammatory M1 and anti-inflammatory M2 macrophages, it is now accepted that these two phenotypes are only the extremes of a continuum of polarization, in which there is a spectrum of differentiated macrophages [60]. Among these differentiated macrophages, TAMs are involved in immune tolerance, inflammatory disease, and cancer [61] and are considered as pro-tumorigenic immune cells. They stimulate Treg differentiation and secrete several factors (e.g. TGFβ, TNFα, and IL-10) to create a favorable environment for tumor progression and to inhibit the anti-tumor effects of immune cells [11]. Regarding prognosis, tumor infiltration by TAMs is reported to be an unfavorable parameter for patient survival [62,63,64]. Our recent study has shown that a high recruitment of CD68+ macrophages in a population of 110 HNSCC was correlated with a shorter patient RFS and OS. Moreover, the analysis of the M1/M2 ratio in the TME, with a double staining using anti-CD68/anti-CD163 antibodies, revealed that 80% of the macrophage population had an M2 phenotype [8]. Interestingly, it appears that TAMs can secrete IL-10 in order to induce the differentiation of T lymphocytes into Treg [65] and thus participate in immune cell evasion.

The existence of complex regulatory loops between these three major mediators of the immune system has led us to quantify their recruitment in ST and IT localizations in a large cohort of head and neck tumors. Indeed, we hypothesized that analyzing each tumor compartment (ST versus IT) may provide distinct and complementary prognostic information. This was also supported in the context of rectal cancer where the location of CD8+ T cells and FoxP3+ Treg cells in distinct compartments (epithelium versus stroma) result in different prognostic responses [66]. Combining the three markers in an IS signature, we found that a low IS was significantly associated with a longer RFS and a prolonged OS. Based on the calculated optimal cut-offs, the IT immune infiltrations associated with a better prognosis correspond to a high density of CD8 and FoxP3 and a low density of CD68 macrophages. Conversely, in the ST, a better prognosis was observed in patients with a low CD8 infiltration but always a high density of FoxP3. Because of the tumor lysis capacity of CD8 cells, these anticancer actors are strong allies for cancer patients. On the other hand, despite their antitumor response suppressor characteristics, Tregs infiltrates have been found to be associated with a favorable outcome, which may be partially attributed to a downregulation of the inflammatory process [44,53]. A correlation had also been found between a higher density of Treg in the stroma and an absence of metastatic lymph nodes, which means that Treg could generate pro-inflammatory processes in the tumor microenvironment favoring a delay in the tumor evolution and consequently would generate a better prognosis of the patients [55]. Furthermore, Khoury et al. recently proposed that IT TILs were distinct from ST TILs in their biological behavior [67]. Indeed, ST is constituted of many components that can impair host immune responses, such as fibroblasts, macrophages, or endothelial cells, underlining the difference in CD8 TILs located in the ST from CD8+ TILs within tumor cells.

Finally, our multivariate analysis, evaluating the tumor stage, the histological grade, and the IS regarding OS, revealed an impressive HR for IS that can significantly predict the OS of patients. By contrast, in head and neck cancer, only a few studies have identified an IS positively correlating with RFS and OS. In two studies, IS only included TILs, and more precisely CD3+ and CD8+ cells [14,29]. A recent study confirmed the positive prognostic impact of TILs in oral squamous cell carcinomas by increasing the immune signature to seven markers that can predict patient survival [68].

## 5. Conclusions

In conclusion, we have shown for the first time that the differential density and local distribution of CD8+, FoxP3+, and CD68+ cells associated in an IS can identify patients with a longer RFS and OS with a stronger significance and a stronger discrimination than TNM classification. Our IS represents an efficient and independent prognostic signature that could constitute a novel indicator beyond TNM staging and histological grade to improve or complement the prediction of clinical outcomes in head and neck cancer patients. Nevertheless, it seems important to note that further studies should be conducted on additional cohorts to validate this promising new IS and to evaluate how it can be applied to predict the response to treatment in head and neck cancer patients.

## Figures and Tables

**Figure 1 cells-11-02050-f001:**
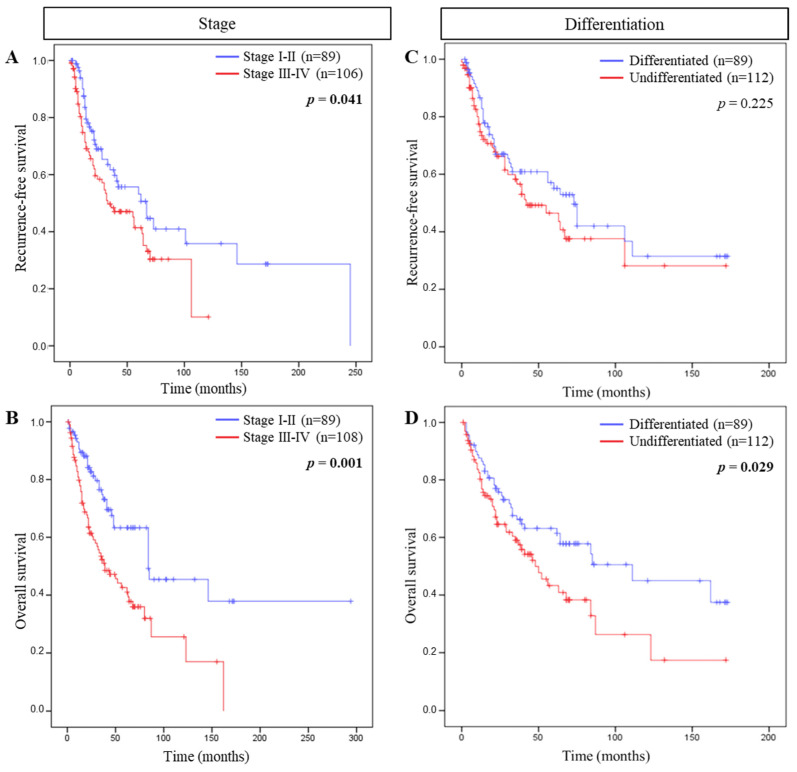
Association between tumor stage and differentiation and patient survival. Kaplan–Meier curves comparing (**A**) recurrence-free survival (RFS) and (**B**) overall survival (OS) of tumor stage and (**C**) RFS and (**D**) OS of tumor differentiation.

**Figure 2 cells-11-02050-f002:**
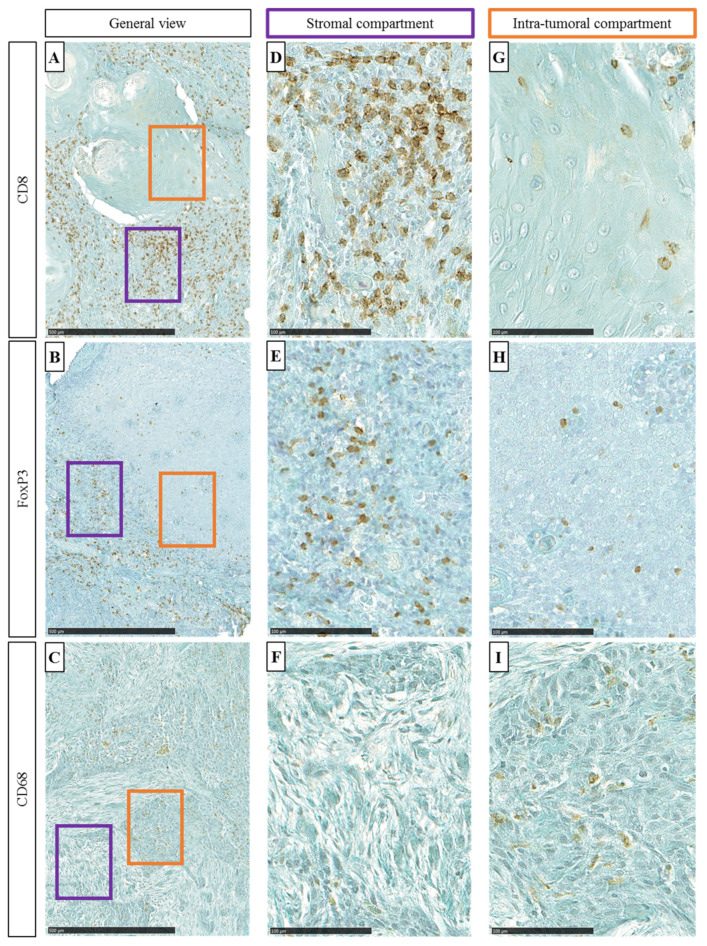
((**A**–**C**)**,** respectively) General view of CD8, FoxP3, and CD68 immunohistochemical staining (100×, scale = 500 µm). Purple oblong represents stromal (ST) area and orange oblong represents intra-tumoral (IT) area. ((**D**–**F**), respectively) Representative images of CD8, FoxP3, and CD68 immuno-marker (400×, scale = 100 µm) in the ST and ((**G**–**I**)**,** respectively) CD8, FoxP3, and CD68 immuno-marker in the IT.

**Figure 3 cells-11-02050-f003:**
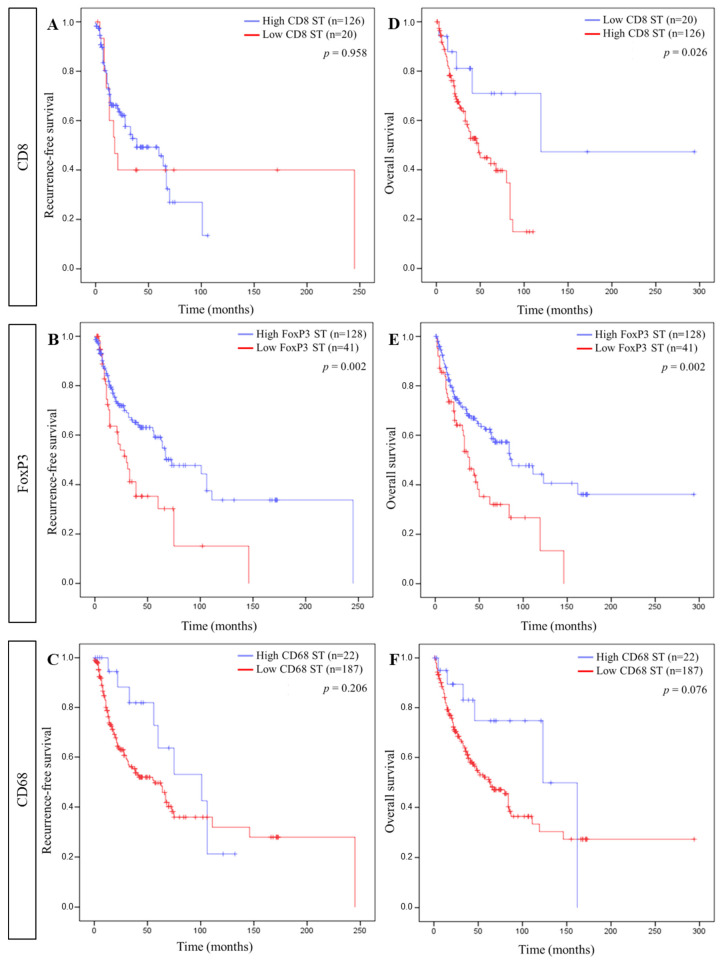
Association between stromal immune cells and patient survival. Kaplan–Meier curves comparing ((**A**–**C**), respectively) recurrence-free survival (RFS) and CD8, FoxP3, and CD68 and ((**D**–**F**), respectively) overall survival (OS) and CD8, FoxP3, and CD68 in the stromal compartment (ST).

**Figure 4 cells-11-02050-f004:**
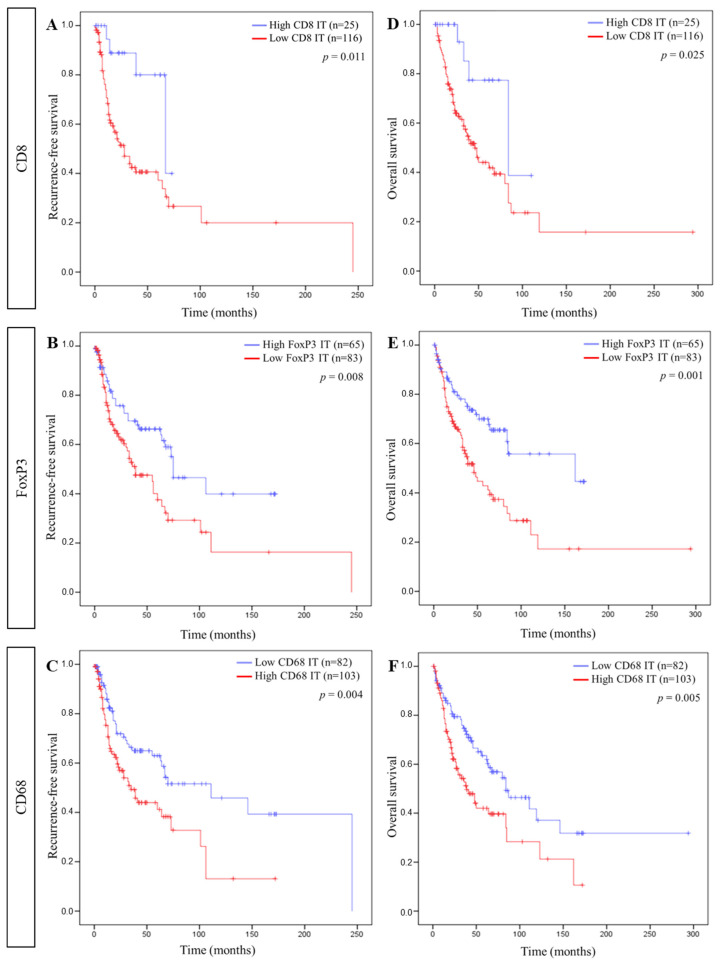
Association between intra-tumoral immune cells and patient survival. Kaplan–Meier curves comparing ((**A**–**C**), respectively) recurrence-free survival (RFS) and CD8, FoxP3, and CD68 and ((**D**–**F**), respectively) overall survival (OS) and CD8, FoxP3, and CD68 in the intra-tumoral compartment (IT).

**Figure 5 cells-11-02050-f005:**
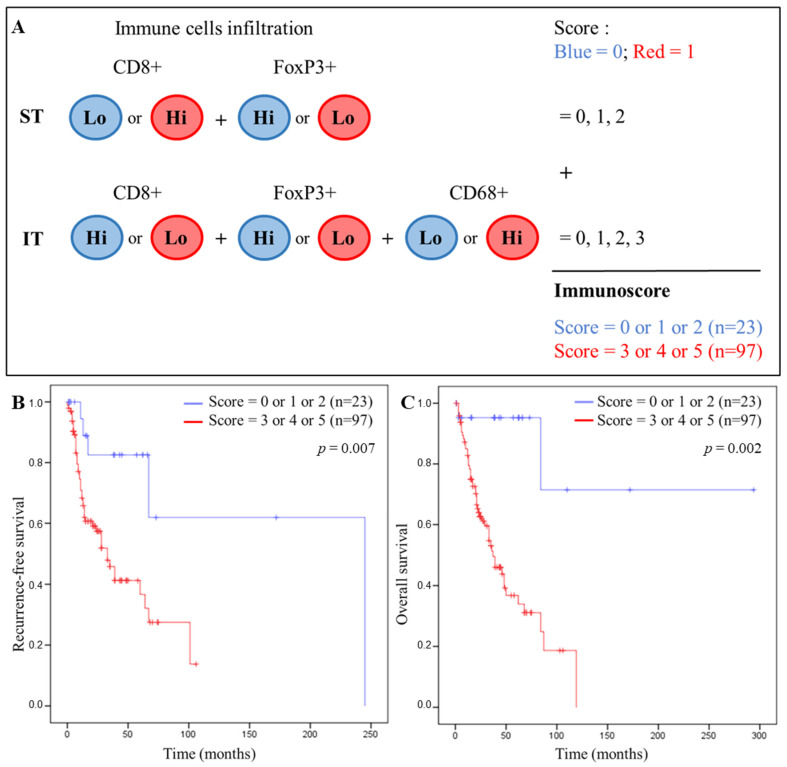
IS and patient survival. (**A**) Establishment of the IS in HNSCC tissues based on the immune cells infiltration in the ST and IT compartment. Each tumor is categorized into low (Lo) or high (Hi) density for each immune cell in each tumor region according to the calculated cutoff values. Depending on the immune cells and the tumor compartment, the Lo and Hi classes are associated to the blue or red group that correspond to 0 or 1 score, respectively. According to the total number of the score, each patient is classified in the blue group (low IS) or the red group (high IS). (**B**) Kaplan–Meier curves comparing recurrence-free survival (RFS) and IS and (**C**) overall survival (OS) and IS.

**Table 1 cells-11-02050-t001:** Patient population characteristics.

Variables	Number of Cases	Relapse-Free Survival		Overall Survival	
	n = 258	*p*-Value	HR (95% CI)	*p*-Value	HR (95% CI)
** Age (years) **					
Median (range)	61 (29–90)				
** Recurrence-free survival (RFS) (months) **					
Median (range)	22 (1–245)				
Yes	104				
No	120				
Unknown	34				
** Overall survival (OS) (months) **					
Median (range)	33 (1–294)				
Alive	124				
Dead	102				
Unknown	32				
** Gender **		**0.048** ^#^	0.67 (0.45–0.99)	0.943	0.98 (0.66–1.48)
Male	177				
Female	81				
** Anatomic site **					
Oral cavity	113				
Oropharynx	80				
Larynx	44				
Hypopharynx	19				
Nasopharynx	2				
** Tumor stage **		**0.041**	1.548 (1.018–2.353)	0.001	2.175 (1.404–3.371)
I-II	84				
III-IV	130				
Unknown	44				
** Histological grade **		0.225	0.76 (0.45–0.99)	0.029	0.62 (0.40–0.95)
Poorly differentiated	112				
Well differentiated	89				
Unknown	57				
** Treatment **					
Surgery	73				
Radiotherapy	14				
Combination surgery and chemo-radiotherapy	28				
Unknown	142				
** Risk factors **					
** *Tobacco* **		0.326	1.32(0.76–2.29)	0.366	1.29 (0.74–2.24)
Smoker	181				
Non-Smoker	36				
Unknown	41				
** *Alcohol* **		0.811	1.05 (0.69–1.60)	0.445	1.18 (0.77–1.81)
Drinker	129				
Non-Drinker	78				
Unknown	51				
** *HPV status* **		0.131	0.65 (0.37–1.14)	0.562	0.86 (0.51–1.44)
Positive	65				
Negative	138				
Unknown	55				
** *p16 status* **		0.103	0.57 (0.29–1.12)	0.152	0.62 (0.32–1.19)
Positive	37				
Negative	121				
Unknown	100				

^#^ Bold *p*-values indicated significant correlations (Cox regression).

**Table 2 cells-11-02050-t002:** Univariate and multivariate Cox regression models evaluating the influence of stromal (ST) or intra-tumoral (IT) of CD8, FoxP3, and CD68 on RFS and OS. Zero and one are related to the cutoff.

Univariate Analysis	Relapse-Free Survival		Overall Survival	
	*p*-Value	HR (95% CI)	*p*-Value	HR (95% CI)
CD8 ST 0-1 ^$^	0.958	0.98 (0.48–2.02)	**0.026** ^#^	3.19 (1.15–8.90)
CD8 IT 1-0	**0.011**	3.73 (1.35–10.31)	**0.025**	3.20 (1.16–8.84)
FoxP3 ST 1-0	**0.002**	1.97 (1.28–3.04)	**0.002**	1.95 (1.29–2.96)
FoxP3 IT 1-0	**0.008**	1.84 (1.17–2.87)	**0.001**	2.16 (1.36–3.41)
CD68 ST 1-0	0.206	1.56 (0.78–3.11)	0.076	2.12 (0.93–4.85)
CD68 IT 0-1	**0.004**	1.86 (1.21–2.85)	**0.005**	1.79 (1.19–2.69)
**Multivariate Analysis**	**Relapse-Free Survival**		**Overall Survival**	
	***p*-Value**	**HR (95% CI)**	***p*-Value**	**HR (95% CI)**
CD8 ST 0-1	0.847	1.10 (0.43–2.78)	**0.028**	5.03 (1.19–21.31)
CD8 IT 1-0	**0.024**	3.36 (1.17–9.67)	**0.038**	3.08 (1.06–8.94)
FoxP3 ST 1-0	0.255	1.87 (0.64–5.49)	0.147	2.46 (0.73–8.28)
FoxP3 IT 1-0	0.056	1.88 (0.98–3.57)	0.109	1.63 (0.90–2.98)
CD68 ST 1-0	0.13	1.63 (0.86–3.09)	0.961	1.02 (0.56–1.83)
CD68 IT 0-1	0.6	1.22 (0.58–2.55)	0.080	1.98 (0.92–4.27)

^$^ Zero and one are groups related to the cut-offs (see Figure 3 and Figure 4). ^#^ Bold *p*-values indicated significant correlations (Cox regression).

**Table 3 cells-11-02050-t003:** Univariate and multivariate Cox regression models evaluating the tumor stage, the histological grade and the IS and the OS.

Univariate Analysis	Overall Survival	
	*p*-Value	HR (95% CI)
Tumor stage	**0.005** ^#^	**1.91 (1.22–3.00)**
Histological grade	**0.029**	0.62 (0.40–0.95)
**ImmuneScore**	**0.002**	9.87 (2.38–40.99)
**Multivariate Analysis**	**Overall Survival**	
	***p*-Value**	**HR (95% CI)**
Tumor stage	0.373	1.36 (0.69–2.69)
Histological grade	0.401	1.31 (0.70–2.46)
**ImmuneScore**	**0.018**	11.17 (1.52–82.12)

^#^ Bold *p*-values indicated significant correlations (Cox regression).

## Data Availability

Data is contained within the article or Appendix A.

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
