# Peer review of "Immunoscore Combining CD8, FoxP3, and CD68-Positive Cells Density and Distribution Predicts the Prognosis of Head and Neck Cancer Patients"

_cells, 2022, doi:10.3390/cells11132050_

Round 1

Reviewer 1 Report

Furgiuele et al. assessed immune cell infiltrate to develop an immunoscore for prognosis and to investigate its correlation with clinical data of patients with head and neck cancer. They found that the differential density and local distribution of CD8+, FoxP3+ and CD68+ cells predict recurrence-free survival and overall survival. They concluded that CD8, FoxP3 and CD68 immunoscore was a strong, independent, and significant prognostic marker that can improve the clinical management of head and neck cancer patients. The work is interesting and could be an important tool for predictive clinical outcomes.

1.    Can the authors take advantage of the publicly available database and confirm their findings using TCGA?

Reviewer 2 Report

In this work the authors analyze the infiltration (both stromal and intratumoral) by CD8+ and FOXP3+ lymphocytes and CD68+ macrophages in a series of 258 head and neck squamous cell carcinoma specimens. The prognostic value of the density of infiltration by these immune cells individually and by combining them in an immunoscore is studied.

Although the subject is not novel, since the prognostic value of infiltration by these cells in head and neck carcinomas has already been demonstrated in multiple previous studies, it does provide the novelty of combining them in an immunoscore. The design of immunoscores that provide added prognostic value to TNM is a current issue, so in this sense it would be a relevant work, but it presents several aspects that need to be clarified/reviewed.

Major points.

- It is a large series of patients, but very heterogeneous, as it includes very varied locations (including two nasopharyngeal tumors) and HPV+ and HPV- tumors. Both the location of the tumor and the relationship with HPV can influence the prognostic value of the infiltration by the cells studied, so a more homogeneous population would have been desirable to avoid this possible source of bias.

-The treatment the patients received is not mentioned, and the type of treatment (surgery, radiotherapy, chemoradiotherapy, combined treatments…) may also affect the prognostic value of infiltration by immune cells. Ideally, all patients should have been treated homogeneously, but in any case, the treatments performed should be indicated, and if possible, an analysis should be made according to the type of treatment.

- As for the determination of HPV, where it has really been shown to have prognostic value and relationship with the immune infiltrate is in oropharyngeal carcinomas, in which it is necessary to have this information. However, although they say in the text (methods) that HPV status was determined in all patients by qPCR (and then checked for transcriptional activity by immunohistochemical expression of p16), Table 1 shows that there were no PCR data in 55 patients and no p16 expression data in 100 patients. The locations in which there were no data are not indicated but should be specified in those of oropharynx.

- The cut-off points obtained to consider low and high infiltration are for this specific series of patients, and should be validated in another series of patients, other locations, treatments... There is still a long way to go to consider the immunoscore obtained applicable.

- It would be very interesting to show if there was a correlation between the infiltration density of the different cell populations studied.

Minor points

- In Table 1, what do you mean by tumor invasion? All carcinomas are assumed to be invasive by definition

- It is surprising that there is no difference in RFS between stage I-II and III-IV cases. To what do you attribute this unexpected result?

- It is also an unexpected result, given the consistency in the previous literature of the opposite result, the association of a low infiltration by CD8+ cells in the stromal compartment with a better prognosis. Even more so taking into account that the opposite happens in the intratumoral compartment. In relation to this, there is an error in line 189, since it indicates that the low intratumoral density of CD8+ lymphocytes is associated with longer RFS and OS, when it is the other way around.

- Some contradictory arguments are raised in the discussion. Thus, in the second paragraph, the prognostic value of infiltration by FOXP3+ lymphocytes is discussed, raising arguments as to why they can be associated with a better prognosis, when given their function, the opposite would be expected. However, in the following paragraph, the negative prognostic value of infiltration by CD68+ cells (macrophages) is argued and one of the mechanisms proposed is that they increase infiltration by FOXP3 lymphocytes, which would participate in immune evasion. So the FOXP3 lymphocytes would have a favorable and detrimental role at the same time.

Round 2

Reviewer 2 Report

The authors have answered all the questions raised by the reviewer, and therefore the work is now acceptable